# An Evaluation of the Nutri-Score System along the Reasoning for Scientific Substantiation of Health Claims in the EU—A Narrative Review

**DOI:** 10.3390/foods11162426

**Published:** 2022-08-12

**Authors:** Stephan Peters, Hans Verhagen

**Affiliations:** 1Nederlandse Zuivel Organisatie (NZO)/Dutch Dairy Association, 2596 BC The Hague, The Netherlands; 2Food Safety & Nutrition Consultancy, 3703 EE Zeist, The Netherlands; 3National Food Institute, Technical University of Denmark, Kemitorvet 201, 2800 Kgs. Lyngby, Denmark; 4Nutrition Innovation Center for Food and Health (NICHE), University of Ulster, Coleraine BT52 1SA, UK

**Keywords:** Nutri-Score, front-of-pack label, health claim, scientific substantiation, EU Regulation 1924/2006

## Abstract

In this narrative review, the scientific evidence in support of the front-of-pack label (FOPL) Nutri-Score system is evaluated along with the reasoning for scientific substantiation of health claims in the EU. A health claim could be phrased as ‘Nutri-Score as an FOPL system results in an increased purchase of healthier foods by consumers’. Peer-reviewed scientific literature as found in Pubmed under search terms ”NutriScore” and “Nutri-Score” that investigate the effects of the Nutri-Score on food purchases were evaluated. In total, eight papers were identified. Only three studies were conducted in real-life settings, and five were on online purchases. In the EU, health claims are evaluated by the European Food Safety Authority (EFSA). Considering the three basic questions that EFSA uses to evaluate scientific substantiation of health claims, it appears that the (i) food/constituent (the Nutri-Score system) is sufficiently defined/characterised, and (ii) the evidence is sufficient to appraise the system as ‘beneficial to human health’. However, the scientific evidence for a (iii) cause-and-effect relationship is contradictory and limited. In conclusion, based on the EFSA approach for substantiation of health claims, there is insufficient evidence to support a health claim based on the Nutri-Score system, since a cause-and-effect relationship could not be established.

## 1. Introduction

The primary goal of a front-of pack label (FOPL) is to stimulate consumers to make healthier food purchases. The efficacy of different FOPLs was recently evaluated in three systematic reviews, including a meta-analysis [1,2,3]. Overall, the studies did report some beneficial results of FOPLs on food choices and dietary intake. However, the individual studies show rather heterogeneous results. The gold standard to evaluate the efficacy of FOPLs is to establish their effect on the composition of the consumer’s actual purchases, e.g., supermarket baskets. One of these systematic reviews [3] addressed the effect of FOPLs on real food purchases. That review could include only 15 studies: 10 randomised controlled trials, 4 pre-post studies, and 1 case-control study. Five studies were conducted in a controlled setting (an online virtual supermarket or physical laboratory grocery store). The other 10 studies were conducted in a naturalistic setting where people commonly purchase foods: supermarket, grocery store, school or hospital cafeteria, and a vending machine. The evaluated FOPLs included traffic lights, health star rating, daily intake guides, health warnings, and high sugar symbol labels. Only 5 of the 12 studies that assessed traffic light labels resulted in participants making healthier food purchase decisions. In addition, one of the two studies that assessed health warning labels and one study that assessed high sugar symbol labels resulted in healthier food purchase decisions. None of the other studies revealed an effect on food purchases compared with controls. The authors concluded that “Findings on the efficacy of FOP nutrition labels in ‘nudging’ consumers toward healthier food purchases remain mixed and inconclusive” [3]. Among FOPLs, the Nutri-Score recently attracted more attention because of its envisaged use in several EU member states [4]. The Nutri-Score FOPL was not evaluated in the abovementioned systematic review [3] because it was not possible to identify the effect of Nutri-Score labels on actual food purchases in a supermarket in PubMed indexed peer-reviewed papers [3]. The added value of this narrative review is that the Nutri-Score is specifically evaluated regarding its efficacy on food purchases.

The basis of all FOPLs are nutrient profiles. Nutrient profiling systems (NPSs) are a way to help communicate health characteristics of foods. In the EU, NPSs are part of Regulation 1924/2006 (on nutrition and health claims on foods). Its Article 4.1 cites,

“the Commission shall […] establish specific nutrient profiles, including exemptions, which food or certain categories of food must comply with in order to bear nutrition or health claims and the conditions for the use of nutrition or health claims for foods or categories of foods with respect to the nutrient profiles”.

In 2008 and again in 2022, the European Food Safety Authority (EFSA) published its Scientific Advice for the setting of nutrient profiles to prepare for the use of health claims on food labels. However, EFSA did not arrive at a definite nutrient profiling system, because “… The European Commission as risk manager will propose the nutrient profiling model… It is not a task for EFSA” (https://www.efsa.europa.eu/en/news/nutrient-profiling-scientific-advice-eu-farm-fork-initiative, accessed on 7 July 2022). Nutrition claims refer to what a food contains and comprise content claims and comparative claims. Health claims refer to what a food does and involve general function claims, claims related to a reduction of risk of disease, and claims related to the growth and development of children. A nutrient profiling logo/FOPL is essentially a combination of a nutrient claim and a health claim. Under EU Regulation 1924/2006, such claims on functional foods must be scientifically substantiated. When it comes to health claims for Europe submitted under Regulation 1924/2006, EFSA provides the scientific advice to the European Commission. To date, EFSA has evaluated over 3000 health claims. The outcome of the scientific evaluation process was that about 250 health claims have been evaluated with a positive outcome [5,6,7].

A relatively new FOPL is Nutri-Score, also known as the ‘5 Colour Nutrition Label’. Nutri-Score is an FOPL of French origin and is currently endorsed by the French Santé Publique [8]. It displays five boxes with colours and letters to grade the nutritional quality of foods and beverages (alcoholic beverages are excluded). This scoring method is a multi-nutrient algorithm based on the UK Food Standard Agency nutrient profiling system (FSA-NPS) [9]. Depending on the end score of the algorithm, healthier foods get an A or green score, and the unhealthiest foods get an E or red score. This is called an across-the-board algorithm, meaning that one set of criteria is applied to all pre-packaged foods, albeit with some small adaptations made for cheeses, fats, and beverages [8]. The Nutri-Score FOPL has been adopted in France, Germany, Luxembourg, Switzerland, and Belgium, and the Netherlands and Spain are in the process of making a choice to adopt it shortly.

Although the Nutri-Score system seems to be gaining political attention and is being promoted for wider introduction and eventually as a candidate for use as a harmonised EU-wide FOPL, appropriate scientific substantiation is required about its use resulting in acknowledged public health benefits. For an FOPL, this would mean that the algorithm is scientifically robust and its efficacy on consumers is scientifically substantiated. Since that claimed effect of the Nutri-Score is on human health, the Nutri-Score system should be evaluated from an obvious angle: via the EU Regulation on Nutrition and Health claims EU 1924/2006 (https://eur-lex.europa.eu/legal-content/en/ALL/?uri=CELEX%3A32006R1924, accessed on 7 July 2022). A tentative health claim for the Nutri-Score system could be phrased as ‘Nutri-Score as an FOPL system results in an increased purchase of healthier foods by consumers’. Healthier food purchases can be defined as improvements of the FSA-NPS score of food purchases. Such a claim would implicitly improve public health too. This paper intends to review the scientific evidence for substantiation of the use of the Nutri-Score system along the lines of scientific evaluation for health claims on food as developed and presented by EFSA (https://www.efsa.europa.eu/en/topics/topic/health-claims, accessed on 7 July 2022).

When evaluating a health claim dossier, EFSA answers three questions, all of which must be satisfactorily answered to allow the agency to reach the conclusion that it is ‘sufficiently scientifically substantiated’. To this end, EFSA evaluates the extent to which:

Q1. The food/constituent is defined/characterised;

Q2. The claimed effect is ‘beneficial to human health’;

Q3. A cause-and-effect relationship is established.

In this paper, all three steps will be evaluated based on the available scientific literature.

## 2. Materials and Methods

This narrative review is based on a selection of published peer-reviewed papers. To this end, the PubMed database was searched on 31 January 2021 with the keywords [Nutri-Score; 119 results] and [NutriScore; 102 results]. From these identified studies, those were selected that evaluated the efficacy of the Nutri-Score with or without other FOPLs on consumer food purchases, food choices, attitude, opinions and trust, etc., resulting in a final set of 18 papers. Out of those remaining 18 studies, 12 were from the same research group or affiliated with the developers of the Nutri-Score, whereas only 6 were conducted by other scientists. These 18 selected studies were divided into three categories:

*RCTs investigating effects of the Nutri-Score on food purchases*. These eight papers are all pertinent to the health claim under investigation. Three of them are by authors affiliated with the developers of the Nutri-Score. These papers were included for evaluation in this review.

*Studies on consumer understanding of the Nutri-Score versus other FOPLs*. These seven papers are relevant but not considered sufficiently pertinent to the health claim under investigation because these consumer-understanding studies only investigated the ability of consumers to use The Nutri-Score in an online setting to score three products in three product groups—cookies, pizzas, and cereals. The studies were neither conducted in a real-life setting nor in a complete supermarket assortment and are therefore not included for evaluation in the present study. All these seven papers are by authors affiliated with the developers of the Nutri-Score.

*Studies on consumer trust and attitude versus the Nutri-Score and other FOPLs*. These three papers are relevant but not considered sufficiently pertinent to the health claim under investigation because they do not investigate the effect of the Nutri-Score on supermarket purchases. Two of these studies are by authors affiliated with the developers of the Nutri-Score.

Hence, only eight papers were RCTs studying the effects of the Nutri-Score on (real) food purchases and were identified as pertinent to the health claim under investigation (‘Nutri-Score as an FOPL system results in an increased purchase of healthier foods by consumers’). A detailed description of the studies that were eligible, relevant, and pertinent and included in this analysis is provided in Table 1.

Of the eight selected papers, five were conducted by researchers not affiliated with the developers of the Nutri-Score. Only three of them were studies that evaluated the efficacy of the Nutri-Score in a real-life setting, namely, a university cafeteria, a real-life grocery store, or an experiment in major chain supermarkets [10]. The other five studies were conducted via online tools. On the basis of the eight pertinent papers selected, the three basic questions for evaluation of the scientific support for the tentative health claim were investigated.

## 3. Results and Discussion

When evaluating a health claim dossier, EFSA answers three questions, i.e., the extent to which [11]:-The food/constituent is characterised;-The claimed effect is ‘beneficial to human health’ (i.e., the context in which a food constituent should be characterised in relation to the claimed effect);-A cause-and-effect relationship is established (i.e., there must be sufficient scientific evidence for the health claim).

These three requirements are discussed below in light of the tentative health claim ‘Nutri-Score as an FOPL system results in an increased purchase of healthier foods by consumers’. Only when all these three requirements are met, can a ‘scientifically sufficiently substantiated’ overall judgement be given. Hypothetically, this could follow under EU provisions in the actual permission for such a health claim. However, if even one such requirement is not fulfilled, EFSA would conclude that the claim is not scientifically substantiated, and consequently the EC would not authorise the health claim. We are aware that such a health claim has not been applied for, but obviously the scientific evaluation can still be performed. This is outlined below in more detail.

### 3.1. The ‘Food/Constituent’ Is Defined/Characterised

The calculation method of the Nutri-Score originates in and is adapted from the UK Food Standards Agency nutrient profiling system (FSA-NPS). A score is calculated per 100 gr or 100 mL of a product based on an algorithm that includes positive or negative points for energy (0–10 points), saturated fats (0–10 points), total sugar (0–10 points), sodium (0–10 points), fibres (−5–0 points), and percentage fruits, vegetables, pulses, nuts and rapeseed, walnut and olive oils (−5–0 points), and proteins (−5–0 points). The total score thus ranges from −15 points for the healthier foods and +40 for the less healthy foods. The foods are then given a designation ranging from A (for the healthier foods) through E (for the less healthy foods). Which final designation the product receives (A–E) depends on the specific upper and lower bounds defined for each of the letters. One set of criteria is used for all pre-packaged foods, with some adaptations for cheeses, fats, and beverages [8,9,12]. The description of the Nutri-Score algorithm is clear and can easily be applied and reproduced. Hence the food/constituent can be interpreted as ‘sufficiently defined’ for usage in an EFSA Health Claim dossier.

### 3.2. The Claimed Effect Is ‘Beneficial to Human Health’

Regulation 1924/2006 describes two categories of claims on foods: nutrition claims and health claims. Nutrition claims refer to what a food ‘contains’: content claims and comparative claims. Health claims refer to what a food ‘does’ in terms of general function claims, those related to a reduced risk of disease and claims related to the growth and development of children. The second step in an EFSA evaluation is to determine whether “the claimed effect is beneficial to human health”. For the Nutri-Score, no such claims have been investigated under the health claims regulation, and so far, no health claim has been approved based on the combined action of nutrients on health. We could not find articles substantiating synergistic or additional health effects of individual nutrients either. Based on this lack of substantiation of the synergistic effects of individual nutrients, the conclusion could already be ‘inconclusive evidence’.

Nevertheless, the theoretical health effect of the Nutri-Score has been estimated based on the application of the FSA-NPS Nutri-Score system in different epidemiological cohorts. In these attempts, estimates have been made of the potential of the Nutri-Score to decrease the incidence of NCDs and mortality in different cohorts. Based on post hoc analyses, the following conclusions have been made:-Consumption of foods with higher FSA-NPS scores is associated with an increased risk of mortality from cancer and cardiocirculatory, gastrointestinal, and respiratory diseases [13].-Consumption of foods with a higher FSA-NPS score is associated with poorer oral health [14].-Consumption of foods with a higher FSA-NPS score is associated with greater asthmatic airway symptoms [15].

Based on the associations found between a poorer Nutri-Score and an increased risk of mortality and morbidity, it could be concluded that the Nutri-Score is potentially beneficial to human health, as better adherence to the FSA-NPS score is associated with a decreased risk. Hence this second requirement—better adherence to the FSA-NPS—can be considered as sufficiently substantiated.

### 3.3. A Cause-and-Effect Relationship Is Established

The Nutri-Score health effects described above are potential and theoretical. The health effects can only be attained if consumers are indeed changing their purchases in such a way that improvement on the FSA-NPS can be observed. The proof of the pudding is in the eating, so the effect of the full-colour Nutri-Score FOPL should be tested on actual, real-life supermarket purchases. However, we did not find publications available to that end. Nor did we find studies that calculated the effects of the Nutri-Score in the selected studies on the FSA-NPS. Therefore, as second choice, we retrieved other studies that evaluated the effects of the Nutri-Score on purchasing behaviour in other settings and using other study designs. A detailed description of the studies that were eligible, relevant, and pertinent and included for this analysis are captured below and summarised in Table 1. The Table identifies the countries in which the study was conducted, a short description of the study and the main results, some critical notes, and whether the authors are affiliated with the developers of the Nutri-Score, are also included.

**Table 1 foods-11-02426-t001:** RCT studies investigating effects of the Nutri-Score on food purchases.

Year of Publication	Country	Authors Affiliated w Nutri-Score	Study Description	Results	Notes	Effect on FSA-NPS	Reference
2019	CO	no	Randomised field trial in a university cafeteria, with randomly provided information on the Nutri-Score. *n*= 484 participants.	Using the Nutri-Score led to more protein, more calories and more expenditures (on healthy items only) in purchases. Purchases of ‘unhealthy’ products did not decrease.	Customers were 10% more likely to buy a healthier item than controls. Information on the Nutri-Score system increased the store’s sales.	Not investigated	Mora-Garcia et al., 2019 [16]
2019	SG	no	RCT investigating the effect of the Nutri-Score compared to the UK’s multiple traffic light system (MTL) and no label in online grocery store. *n* = 154 participants in a 3 × 3 crossover (within-person) design.	The Nutri-Score and MTL performed significantly better vs. no-label controls. NS performed statistically better than MTL and control-based on average Nutri-Score. MTL (but not the Nutri-Score) statistically reduced calories or sugar from beverages.	Thorough study with crossover design.	Not investigated	Finkelstein et al., 2019 [17]
2021	BE	no	A difference-in-difference analysis of a natural experiment in 43 supermarkets of a major retailer in Belgium versus 14 control stores, studying the impact of shelf tags with the Nutri-Score on consumer purchases.	The proportion of Nutri-Score B and C product sales was more favourable in intervention than control stores and less favourable for Nutri-Score D product sales. A positive impact was found for 17/58 food categories (vegetable, fruit and dairy products, and confectionery), a negative impact for 16/58 categories (bread and bakery products).	The impact on consumer purchases was mixed as difference–in-differences found were favourable for Nutri-Score B and C products and unfavourable for Nutri-Score D products. Shelf labelling on its own is unlikely to significantly influence consumer behaviour.	Not investigated	Vandevijvere and Berger 2021 [10]
2020	FR	no	RCT investigating four FOPLs (SENS, Nutri-Score, Nutri Repère, Nutri-Couleurs) to improve the nutritional quality of food purchases in real-life grocery shopping settings.	The Nutri-Score increased purchases of foods in the top-third of their category by 14% nutrition-wise, but had no impact on purchases of foods with medium, low or unlabelled nutritional quality.	The Nutri-Score improved the nutritional quality of labelled foods purchased by only 2.5% in the FSA-NPS score. Effect sizes were 17 times smaller on average than those found in comparable laboratory studies.	Yes, but based on four product groups only	Dubois et al., 2020 [18]
2019	FR	yes	Three RTCs in students (*n* = 1866), low-income individuals (*n* = 336), and cardiovascular patients (*n* = 1180) investigating the effect of the Nutri-Score on overall nutritional quality of purchases in an online supermarket compared to the RIs and no label.	Shopping cart contents were lower in calories and saturated fatty acids and higher in fruits and vegetables in the Nutri-Score arm than in the other arms.	No significant difference between the Nutri-Score and no-label groups or between RIs and no-label groups.	Yes, but no significant effect of Nutri-score versus no label	Egnell et al., 2019 [19]
2021	FR	yes	Three RTCs in students (*n* = 1866), low-income individuals (*n* = 336), and cardiovascular patients (*n* = 1180) investigating the effect of the Nutri-Score on purchasing intentions in an online supermarket compared to RIs and no label.	Shopping carts of participants simulating purchases with the Nutri-Score affixed to pre-packaged foods contained a higher proportion of unpacked products—especially raw fruits and meats, i.e., with no FoPL—comparedto participants purchasing with no label or withRIs.	This is a sequel paper to the one above (#17). It is a post-hoc analysis, viz. analyses that were not originally planned.“The Nutri-Score appears to decrease purchases in processed products resulting in higher proportions of unprocessed and unpacked foods, in line with public health recommendations.”	Not investigated	Egnell et al., 2021 [20]
2021	FR	yes	RCT investigating the effect of the Nutri-Scorecompared to RIs and no label. Participants (*n* = 336) went on a simulated grocery shopping at an experimental online supermarket.	The Nutri-Score performed significantly better versus RIs (overall nutritional quality of the shopping cart, and lower caloric and saturated fatty acids content), but not versus no label.	This is one of the three arms in the study mentioned above published as a separate paper.	Yes, but no significant effect of Nutri-score versus no label	Egnell et al., 2021 [21]
2021	NL	no	Investigate the effect of the Nutri-Score on (*n* = 192) consumer attitudes, taste perception, and purchase intention in an online environment by comparing the Nutri-Score’s efficacy on three different snacks labelled with Nutri-Score A, B, and C.	No effects of the Nutri-Score were observed on attitudes, taste perception, or purchase intention.	This study is similar to those conducted by the Nutri-Score’s developers.	Not investigated	Folkvord and Pabian 2021 [22]

BE = Belgium; FR = France; CO = Colombia; NL = the Netherlands; SG = Singapore; FSA-NPS = Food Standard Agency Nutrient Profile Score; RIs = reference intakes.

The health claim under review, ‘Nutri-Score as an FOPL system results in an increased purchase of healthier foods by consumers’, can be investigated in RCTs comparing the Nutri-Score (and other FOPLs) versus no label. RCTs are typically the strongest study designs to investigate the existence of a relationship between exposure and effect [6,23]. In this case, exposure can be understood as seeing the FOPL/Nutri-Score on packaging, and the ultimate effects can be investigated from actual purchasing behaviour. Our literature survey identified only three RCTs evaluating the Nutri-Score system for ‘actual purchases’: a university cafeteria [16], a real-life grocery store [18], and an experiment in major chain supermarkets [10], in addition to several online studies [17,19,20,21,22].

Mora-Garcia et al. [16] conducted a randomised field trial in a university cafeteria with 484 participants, randomly providing information on the Nutri-Score. The use of the Nutri-Score led to more protein, more calories, and higher expenditures (on items with a better Nutri-Score only), while the purchasing of ‘unhealthy’ products did not decrease. Customers were 10% more likely to buy a healthier item than controls. Information on the Nutri-Score system also increased the cafeteria’s sales. However, no calculations have been made on the effects of the Nutri-Score on the FSA-NPS.

Dubois et al. [18] conducted an RCT investigating four FOPL (SENS, Nutri-Score, Nutri Repère, Nutri-Couleurs) to improve the nutritional quality of food purchases in a real-life supermarket setting. The study was peer-reviewed but not indexed in PubMed. Since this is a real-life study using the Nutri-Score, it was decided to include it in our evaluation. FOP labels were placed on products of four categories, namely freshly prepared meals, pastries, bread, and canned/prepared meals. The Nutri-Score increased the purchases of foods in the top third of their category by 14% nutrition-wise but had no impact on the purchases of foods with medium, low, or unlabelled nutritional quality. The Nutri-Score only improved the nutritional quality (measured by FSA-NPS) of purchased labelled foods by 2.5%. This effect was mainly due to the effect of the Nutri-Score on the food group of freshly prepared meals. Although this is a very large study, it only investigated the effects of FOPLs in four food groups on dedicated shelves in different sections of a supermarket. The authors also questioned whether the 2.5% effect of the Nutri-Score on the FSA-NPS score was clinically relevant and noted that effect sizes were 17 times smaller on average than those found in comparable laboratory studies they conducted. This study comes close to studying the effect in a real supermarket. As such, it should be regarded as a field experiment of the effect of the Nutri-Score on four product groups in a real-life supermarket setting.

Vandevijvere and Berger [10] evaluated the impact of black-and-white electronic shelf labels (ESL) with the Nutri-Score on consumer purchases overall and by food category in 43 intervention supermarkets of a major retailer in Belgium versus 14 control stores. Each week, non-food and food sales for 2018 and 2019 were received by the Nutri-Score (A/B/C/D/E) per food category. The primary outcomes were the proportion of food sales for Nutri-Score A/B/C/D/E. Difference-in-differences regression analysis was conducted to estimate the effect of the ESL intervention on the proportion of overall food and food category sales for Nutri-Score A/B/C/D/E. Difference-in-differences for the proportion of Nutri-Score B and C product sales were found to be more favourable in intervention than control stores and less favourable for Nutri-Score D product sales. A positive impact was found for 17/58 food categories (29% of total food sales) and a negative impact for 16/58 categories (24% of total food sales). Positive impacts were found for vegetable, fruit, and dairy products and for confectionery. Negative impacts were found for bread and bakery products. The authors concluded that the impact of ESL on consumer purchases was mixed, as difference-in-differences found were favourable for Nutri-Score B and C products and unfavourable for Nutri-Score D products. The final conclusion was that shelf labelling on its own is unlikely to significantly influence consumer behaviour. This study is the only one to investigate the efficacy of the Nutri-Score on consumers’ purchasing behaviour in a complete supermarket assortment, although it was only applied in black-and-white form on the shelf-labels. The efficacy of the Nutri-Score was only interpreted for the respective Nutri-Scores A–E, and its effect on the FSA-NPS was not calculated.

It should be noted that the three studies described above on actual purchasing behaviour were all conducted by research groups that have no connection with the originators of the Nutri-Score system, so they can be considered fully independent. In addition to real-life experiments, there are also online experimental studies that can be relevant for the evaluation of the defined health claim ‘Nutri-Score as an FOPL system results in an increased purchase of healthier foods by consumers’.

Finkelstein et al. [17] investigated the NS system and MTL on purchases in an online grocery store. They ran a thorough study with crossover design: an RCT investigating the effect of the Nutri-Score compared to the UK’s multiple traffic light system (MTL) and no label. Their study used 154 participants in a 3 × 3 crossover (within-person) design. The Nutri-Score and MTL performed significantly better versus no-label controls based on average Nutri-Score values. However, the effects on the FSA-NPS were not calculated. The Nutri-Score performed statistically better than MTL and controls. By contrast, MTL—but not the Nutri-Score—statistically reduced the intake of calories or sugar from beverages.

The originators of the Nutri-Score system conducted three more RCTs in different populations—students (*n* = 1866), low-income individuals (*n* = 336), and cardiovascular patients (*n* = 1180)—investigating its effect on purchasing intentions in online supermarkets compared to the reference intakes (RIs) and no label. In the original study [19], the shopping cart contents were lower in calories and saturated fatty acids and higher in fruits and vegetables in the Nutri-Score arm than in the other arms. The effects of the Nutri-Score on the FSA-NPS was not significantly different from no label but was better when compared to the RIs. In a sequel analysis to the same three RCTs [20], shopping carts of participants simulating purchases with the Nutri-Score affixed to pre-packaged foods contained a higher proportion of unpacked products—especially raw fruits and meats, i.e., with no FOPL—compared to participants purchasing with no label or with the RIs. In this post hoc analysis (i.e., not originally planned), according to the authors “Nutri-Score appears to decrease purchases in processed products resulting in higher proportions of unprocessed and unpacked foods, in line with public health recommendations.” Whereas a post hoc analysis already tends to have a lower scientific value, this suggests that the Nutri-Score FOPL does not stimulate the purchase of unlabelled (fresh) products over labelled products (with Nutri-Score). One of the arms of the study was also reported by Egnell et al. [21] in an RCT investigating the effect of the Nutri-Score compared to RIs and no label. Participants (among low-income individuals, *n* = 336) went on a simulated grocery shopping at an experimental online supermarket. The Nutri-Score performed significantly better versus RIs (overall nutritional quality of the shopping cart and lower caloric and saturated fatty acids content) yet did not improve statistically significantly versus controls (no label). The ultimate conclusion from this study is not convincing but worded carefully and weakly: “Nutri-Score […] *appears to have the potential* to encourage purchasing intentions of foods from higher nutritional quality compared with the RIs label”. However, in light of the health claims evaluation practices, it would be concluded that there is no effect of the intervention since the Nutri-Score did not perform better than the controls (i.e., no label), and no significant effects were found on the FSA-NPS.

The study of Folkvord et al. [22] is similar to those conducted by Nutri-Score’s developers as discussed above. These authors are not affiliated with the developers of Nutri-Score. They investigated the effect of the Nutri-Score on (*n* = 192) consumers’ attitudes, taste perception, and purchase intention in an online environment by comparing Nutri-Score’s efficacy on three different snacks labelled with Nutri-Score A, B, and C. No effects of the Nutri-Score were observed on attitudes, taste perception, or purchase intention. The effects of the Nutri-Score on the FSA-NPS were not examined.

Whereas studies on consumer understanding of the Nutri-Score versus other FOPLs and studies on consumer trust and attitude versus the Nutri-Score and other FOPLs are not considered sufficiently pertinent to the health claim under evaluation, the limited and meagre evidence they provide on an effect of the Nutri-Score contributes to the notion of insufficient scientific evidence to support potential health effects of the Nutri-Score FOPL. Those 10 studies, 8 of them by scientists associated with the developers of the Nutri-Score, are mentioned in Figure 1.

There are limitations to this study. The literature search has been limited to Pubmed indexed papers, so studies might be missing in the narrative review. The added value of this narrative review is that it evaluated the real-life effect of Nutri-Scores’ effectiveness on food purchases, which has not been carried out before.

In summary, the outcome of the RCTs provides little evidence of the effects of the Nutri-Score in a real supermarket with a complete assortment and only very limited evidence of an effect of the Nutri-Score on purchasing behaviour and food choice in real life and online testing environments.

## 4. Conclusions

Whereas in the EU, health claims need to be substantiated by firm scientific evidence, to date, the evidence supporting the tentative health claim ‘Nutri-Score as an FOPL system results in an increased purchase of healthier foods by consumers’ is at best insufficient. Only the first two of the three boxes that need to be ticked for a health claim to be determined sufficiently substantiated by EFSA is largely undisputed: the system is clearly described and can be readily reproduced by others, and the science supporting the box on being beneficial to human health can be judged overall as sufficiently underpinned. The claimed health effects of the Nutri-Score as supported by epidemiological studies [14,24] and by WHO/IARC [25] are based on the change in FSA-NPS. The third box to be ticked is that of a cause-and-effect relationship between the effect of the Nutri-Score on the FSA-NPS. Only one real-life study has found a small effect of the Nutri-Score of supermarket purchases on the FSA-NPS, but this was only applied for four product groups [18]. No efficacy study has found an effect of the Nutri-Score on the FSA-NPS for a complete supermarket assortment. In conclusion, there is certainly not enough scientific evidence to substantiate a cause-and-effect relationship, as the Nutri-Score shows diverse outcomes in the various studies selected in support of this claim. Even those scientists involved in the development of the Nutri-Score system carefully chose their words on their ‘support’ for this claim.

In summary, there seems to be only inconclusive or insufficient evidence to support a health effect of the Nutri-Score system. The EU has not authorised any health claim based on insufficient and limited scientific evidence. Before introducing any FOPL, it is advised to prove its efficacy in a full assortment in a real-life supermarket. It is also advised that the scientific substantiation be tested by an independent scientific committee. In Europe this role is executed by EFSA.

## Figures and Tables

**Figure 1 foods-11-02426-f001:**
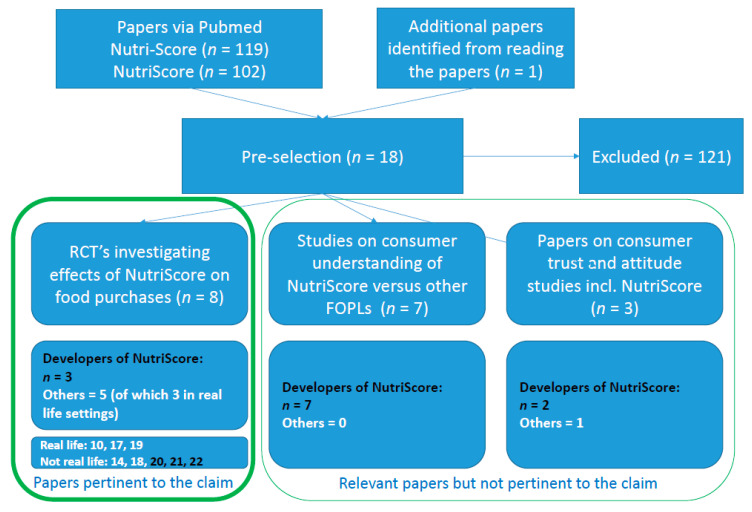
Study selection flow diagram, including the annotation of studies that were or were not conducted by scientists that originally developed the Nutri-Score system. References numbered in bold black are by the developers of the Nutri-Score.

## Data Availability

No new data were created or analyzed in this study. Data sharing is not applicable to this article.

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
