# Peer review of "An Evaluation of the Nutri-Score System along the Reasoning for Scientific Substantiation of Health Claims in the EU—A Narrative Review"

_foods, 2022, doi:10.3390/foods11162426_

Round 1

Reviewer 1 Report

Review Report of the research paper entitled: “An evaluation of the Nutri-Score system along the reasoning for scientific substantiation of health claims in the EU.”

Overall Evaluation

This paper, evaluating the scientific evidence of the Nutri-Score system, is of particular relevance and could add to the literature if published. However, I encourage the authors to perform some major and minor changes.

Abstract

1.     The authors are instructed to follow the journal recommendations regarding the allowed word count. (Please review the authors’ guidelines provided by the journal)

Introduction

2.     Please pay attention to abbreviations. For example, if the phrase "European Food Safety Authority" is abbreviated, there is no need to repeat it multiple times.

3.     Could you explain more about your choice of the following theorized health claim for your investigation: “Nutri-Score as a FOPL system results in an increased purchase of healthier foods by consumers.”? How did you base this choice? Please elaborate more.

Methods

4.     Although your search strategy is justifiable, please elaborate on the eligibility criteria followed to include the papers in your review. Add your exclusion and inclusion criteria under the Methods section. Were there any restrictions to language? Country? Date of study? Study’s type? These need to be elaborated on more in detail. 

5.     I suppose to move Table 1: “RCT studies investigating effects of Nutri-Score on food purchases.” to the Results section, as this table shows the summary of studies included in your review as a result of your search.

Results and Discussion

6.     Please add your study limitations, as well as its strengths. Among your study’s limitations, it is important to mention that some studies might be missing as you searched only one data base.  

Conclusions

7.     I would like to see your future perspectives concerning this study? What do you recommend as a future work?

Tables

8.     Please define country abbreviations under Table 1.  

Author Response

Dear reviewer, please find attached our responses to your review and in track changes all changes we have made based on your and the other reviewers comments.

Reviewer 2 Report

Thank you for giving me the opportunity to review the manuscript entitled „An evaluation of the Nutri-Score system along the reasoning for scientific substantiation of health claims in the EU”.

The paper is informative, clear and in the scope of the Journal.

Please see below my specific comments to the paper:

1.      Information that the paper is narrative review should be added into the title

2.      In the abstract more information should be added about criteria for paper selection: in English, year of publication etc.

3.      Although the introduction part of the paper is rather long I would expect additional sentence explaining the added value of current paper to the three systematic reviews recently published.

4.      I would make the conclusion part of the paper shorter (one paragraph) and more précised – to clearly present the conclusions from the review and recommendations for researchers and society.

Author Response

(The authors gave the same response as above.)
